# Multiple *Clonostachys rosea* UDP-Glycosyltransferases Contribute to the Production of 15-Acetyl-Deoxynivalenol-3-O-Glycoside When Confronted with *Fusarium graminearum*

**DOI:** 10.3390/jof9070723

**Published:** 2023-07-02

**Authors:** Kelly A. Robinson, Antony D. St-Jacques, Sam W. Shields, Amanda Sproule, Zerihun A. Demissie, David P. Overy, Michele C. Loewen

**Affiliations:** 1Aquatic and Crop Resources Development Research Center, National Research Council of Canada, 100 Sussex Drive, Ottawa, ON K1A 0R6, Canada; 2Ottawa Research and Development Center, Agriculture and Agri-Food Canada, Ottawa, ON K1A 0Z2, Canada

**Keywords:** *Clonostachys rosea*, *Fusarium graminearum*, biocontrol, mycotoxin detoxification, deoxynivalenol glycosylation, uridine diphosphate glycosyl transferase

## Abstract

Mycotoxins, derived from toxigenic fungi such as *Fusarium*, *Aspergillus,* and *Penicillium* species have impacted the human food chain for thousands of years. Deoxynivalenol (DON), is a tetracyclic sesquiterpenoid type B trichothecene mycotoxin predominantly produced by *F. culmorum* and *F. graminearum* during the infection of corn, wheat, oats, barley, and rice. Glycosylation of DON is a protective detoxification mechanism employed by plants. More recently, DON glycosylating activity has also been detected in fungal microparasitic (biocontrol) fungal organisms. Here we follow up on the reported conversion of 15-acetyl-DON (15-ADON) into 15-ADON-3-O-glycoside (15-ADON-3G) in *Clonostachys rosea*. Based on the hypothesis that the reaction is likely being carried out by a uridine diphosphate glycosyl transferase (UDP-GTase), we applied a protein structural comparison strategy, leveraging the availability of the crystal structure of rice Os70 to identify a subset of potential *C. rosea* UDP-GTases that might have activity against 15-ADON. Using CRISPR/Cas9 technology, we knocked out several of the selected UDP-GTases in the *C. rosea* strain ACM941. Evaluation of the impact of knockouts on the production of 15-ADON-3G in confrontation assays with *F. graminearum* revealed multiple UDP-GTase enzymes, each contributing partial activities. The relationship between these positive hits and other UDP-GTases in fungal and plant species is discussed.

## 1. Introduction

Mycotoxins, derived from toxigenic fungi such as *Fusarium*, *Aspergillus,* and *Penicillium* species have impacted the human food chain for thousands of years [1]. Such mycotoxins include an array of secondary metabolites such as deoxynivalenol (DON), fumonisins, T-2 toxin, zearalenone, aflatoxins, ochratoxins, and patulin etc. [2]. Their accumulation in agricultural crops, grain stores, and the environment more broadly remain associated with adverse health effect to animals as well as humans upon consumption or inhalation [3,4], most recently including links to cancer [5].

DON, a tetracyclic sesquiterpenoid type B trichothecene is predominantly produced by *F. culmorum* and *F. graminearum* during the infection of corn, wheat, oats, barley, and rice [6,7]. Ingestion of DON through the consumption of infected feed by animals can lead to diarrhea, vomiting and gastroenteritis, arising from the inhibition of protein translation via interference with peptidyl transferase function on the ribosome as well as by modifying gene expression, culminating in cellular death [8]. Various modified analogs of DON are reported, where modifications that include the acetylation, glycosylation, de-epoxidation, epimerization and hydroxylation of DON and DON derivatives are associated with enzymatic activity in plant hosts or by other microbial species that have evolved a protective strategy to detoxify DON [2,7]. With anticipated increases in the spread of toxigenic fungal species due to climate change, understanding the underlying DON detoxification mechanisms has become a priority based on the adverse health effects noted above.

DON detoxification by glycosylation is a protective mechanism commonly observed in host plants. A survey of the current literature highlights examples of DON glycosylation across multiple crop hosts that include maize, barley, rice, oats, wheat, *Aegilops,* and *Arabidopsis* [9,10,11,12,13,14,15]. In a number of these cases, specific enzymes belonging to UDP-Glycosyl Transferases (UDP-GTases) Family 1 class of enzymes were implicated in the production of DON-3-O-glucosides (D3G) and other glycosylated analogs. Some of the UDP-GTases are encoded by the genes AET5Gv20385300 in *Aegilops*, AVESA.00010b.r2.6AG1068650.1 and AVESA.00010b.r2.6AG1068570.1 in oat [13,14], Os04g0206600 (Os70) in rice [16], and include the proteins HvUGT13248 in barley [9,12,17] and UGT73C5 in *Arabidopsis* [11]. Of the fore mentioned enzymes, the rice enzyme Os70 is the most thoroughly characterized, including functional studies based on a 3-dimensional Os70 co-crystal structure with a fluorinated uridine-diphosphate-glucose analogue and bound trichothecenes (PDB ID 5TMD) [16,18]. 

Beyond plants, DON glycosylation activity also occurs in two fungi, both of which are documented *F. graminearum* mycoparasites (and therefore proposed as biocontrol organisms) [19,20]. In plate confrontation assays with *F. graminearum*, a *Trichoderma sp.* was found to detoxify DON into D3G [20], while *C. rosea* converted 15-acetyl-DON into 15-acetyl-DON-3-glucoside (15-ADON-3G) [19]. However, at this time, the identity of the enzymes responsible for the glycosylation activity are unknown, although fungal enzymes associated with the UDP-GTase Family 1 may be responsible.

In the current study, we follow up on the conversion of 15-ADON into 15-ADON-3G in *C. rosea*, based on the hypothesis that the reaction is likely being carried out by a Family 1 UDP-GTase. Due to a lack of amino acid sequence homology between plant and fungal UDP-GTases, we applied a protein structural comparison strategy, leveraging the availability of the crystal structure of rice Os70, to identify a subset of potential *C. rosea* UDP-GTases that might have activity against 15-ADON. Using CRISPR/Cas9 technology, we subsequentially knocked out a selection of target UDP-GTase in the *C. rosea* strain ACM941. Evaluation of the impact of knockouts on the production of 15-ADON-3G in confrontation assays with *F. graminearum*, revealed multiple UDP-GTase enzymes each contributing partial activities. The relationship between these positive hits and other UDP-GTases in fungal and plant species is discussed.

## 2. Materials and Methods

### 2.1. Chemicals

All reagents were purchased from Sigma^TM^, unless otherwise indicated below.

### 2.2. Sequence Data Mining and In Silico Structural Modelling of C. rosea UDP-Glycosyltransferases

The *C. rosea* CBS125111v1.0 genome in the JGI database was subjected to a string search using the terms “glucosyl transferase udp”, “glucosyltransferase udp”, “glycosyltransferase udp”, “glycosyl transferase udp” as well as “lycosyl synthase udp” and “glucosyl synthase udp”. JGI hits that were assessed to belong to the UDP glycosyltransferase Family 1 enzyme group (EC 2.4.1.x) based on sequence homology, were subjected to homology modelling using the SWISS-Model [21]. All figures depicting protein structures or catalytic sites were made using PyMol (Schrödinger).

### 2.3. Fungal Strains, Preparation of Conidia and Protoplast Production

*Clonostachys rosea* (ACM941) was provided from A. Xue (Ottawa Research and Development Centre, Agriculture and Agri-Food Canada (AAFC)) with permission from Adjuvant Plus Inc. *Fusarium graminearum* strain GZ3639 (Proctor et al., 1995) was obtained from S. McCormick (United States Department of Agriculture, Peoria). ACM941 and Fg3639 macroconidia stocks were created and stored at −80 °C as previously described [22], with one additional wash step prior to determining the concentration. 

ACM941 protoplasts were produced as described previously with minor modifications [23]. Two 500 mL flasks both with 100 mL potato dextrose broth (PDB) were inoculated with 500 µL of 2 × 10^7^ ACM941 conidia/mL and grown for 24 h at 125 rpm and 22 °C. Mycelia were harvested, and together washed and combined with the protoplasting solution as described previously. This was incubated at 22 °C with shaking at 50 rpm for approximately 2 h and 45 min, checking the protoplast development after 45 min, 1 h and the final hour. The protoplasts were separated from the mycelia by filtration through Miracloth^®^ followed by washing of the Miracloth^®^ with 25 mL 1.2 M KCl, combining both the filtrates together. The obtained combined filtrate was centrifuged at 2500× *g* for 10 min at 4 °C and the protoplast pellet subsequently washed three times with 30 mL 1.2 M KCl and finally resuspended in 5 mL 1× STC buffer (1.2 M Sorbitol, 10 mM Tris-HCl [pH 8.0], 50 mM CaCl_2_). This yielded a final concentration of 10^7^ to 10^8^ protoplasts per mL which were mixed with 7% DMSO and aliquoted (200 µL) for storage at −80 °C.

### 2.4. CRISPR/Cas9 Knockout of Target C. rosea UDP-Glycosyltransferases

The coding sequences of individual UDP-GTase genes in the *C. rosea* genome were deleted as described previously [23,24]. The selected protospacer sequences were blasted against the *C. rosea* genome to confirm there was less than 15 bp homology between the protospacer and any off-targets in the genome. The in vitro assembly of the Cas9-gRNA ribonucleoprotein (RNP) complexes was performed where the ribonucleoproteins (RNPs), including the target specific crRNA (Appendix A), tracrRNA (trans-activiating CRISPR RNA), and Cas9 nuclease were ordered from: Integrated DNA Technologies (IDT). The selectable marker, hygromycin B (Hyg), was used to replace the UDP-GTase genes of interest with the hygromycin repair template amplified from the Prf-HU2 vector [25] using designed primers (Appendix A) and the Q5^®^ Hot Start High-Fidelity DNA Polymerase enzyme (NEB, M0493), following the thermocycling conditions specified in the manual, with the annealing temperature at 68–69 °C. The amplified templates were PCR purified (QIAGEN) resulting in a Hyg repair template with Hyg flanked by 35–50 bp microhomology regions which target coding regions for the GTase genes. Transformation of the *C. rosea* protoplasts were conducted where 200 µL of protoplasts were mixed with 26.5 µL RNP complex, 8 µg of purified Hyg repair template, 25 µL of polyethylene glycol (PEG)-CaCl_2_ buffer in a 15 mL sterile snap cap tube and incubated for 1 h on ice. A negative control was included that omitted the RNP complex and Hyg repair template. 1.5 mL of PEG-CaCl_2_ buffer was added to the mixture, mixed, and incubated for an addition 20 min. Subsequently, the volume was diluted to 2 mL with 1× STC buffer followed by 2 mL (total volume = 4 mL) of TB3 liquid media and mixed by pipetting. This mixture incubated for 18 to 20 h at 25 °C with 150 rpm shaking. 300 µL of the transformation was mixed with 15–20 mL of TB3 molten (containing low-melting point agar) supplemented with 100 mg/L Hygromycin B (Bioshop Canada Inc.) and poured in petri dishes. Six plates per gene knockout were plated to increase the likelihood and quantity of colonies. The plates were incubated at 28 °C for 2–4 days to allow the mycelium to emerge; upon seeing individual colonies, sterile glass cloning cylinders were used to isolate the colony and allow mycelia to grow. Putative transformants were transferred to potato dextrose agar (PDA) plates with 150 mg/L of Hyg B for another round of selection. An agar plug was used to inoculate 5 mL of PDB with 150 mg/L of Hyg B which was grown for 2 days, filtered through cheesecloth where 20 µL of the filtrate was plated on PDA + Hyg (150 mg/L) plates. Single spore colonies were isolated using cloning cylinders again and transferred to a final selection plate of PDA + 150 mg/L Hyg B. The mycelium growth on this final plate was used to inoculate PDB for genomic DNA isolation, and to inoculate 50 mL CMC media for conidia production and extraction. 

### 2.5. Genomic DNA Isolation and Confirmation of Knockouts Using PCR and Whole Genome Sequencing

Genomic DNA (gDNA) was isolated from mycelia that grew on the sides of 50 mL PDB cultures inoculated with 4 agar plugs of the target strain. The mycelia were collected and frozen using liquid nitrogen and then ground into a powder using a previously chilled mortar and pestle. The gDNA was isolated from the ground mycelia using the E.Z.N.A. ^®^ Fungal DNA Kit (Omega Bio-tek, D3390-01), following the procedure for Fresh or Frozen specimens. DNA quantity and quality was confirmed using a Nanodrop spectrophotometer and by a 0.8% agarose gel. The obtained gDNA was submitted for whole genome sequencing either at the National Research Council of Canada (Saskatoon, Canada) using an Illumina MiSeq with 600 cycle V3 chemistry (2 × 301 paired end) at a sample loading concentration of 12 pM + 1% 12.5 pM PhiX Control using the NEBNext Ultra II FS DNA Library Prep Kit for Illumina (E7805 E6177), or at Genome Quebec (Canada) using a NovaSeq6000 PE150 (35M reads) following an Illumina shotgun library preparation. The obtained data were analyzed as described previously [23], using the Qiagen CLC-Genomics Platform (v.21.0.5) with default settings, but with trimmed sequences aligned to the *C. rosea* CBS125111v1.0 reference genome in the JGI database. 

The initial step involved conducting a quality control check using FastQ Screen software [26] on the raw paired-end sequenced reads to identify contaminants. The FastQC (https://www.bioinformatics.babraham.ac.uk/projects/fastqc/) tool (accessed on 24 October 2022) was then utilized to assess the overall quality of the paired-end reads. Subsequently, TrimGalore (https://www.bioinformatics.babraham.ac.uk/projects/trim_galore/) was applied to remove adaptors and filter out low-quality reads. The parameters utilized in TrimGalore included trimming five base pairs at the 5’ end and one base pair at the 3’ end for read 1 and 2 (--clip_R1 5, --clip_R2 5, --three_prime_clip_R1 1, and --three_prime_clip_R2 1), removing unknown base pairs from both ends (--trim-n), filtering reads with a length of less than 50 bp and a quality score of less than 20 on the Phred scale (--length 50 and -q 20), and eliminating adaptors with an overlap stringency of 5 bp (--stringency 5). After filtering, the quality of the trimmed reads was assessed again using FastQC. The trimmed reads were then aligned to the ACM941 reference genome using the Burrows-Wheeler Aligner (BWA) software in the paired-end mode of the BWA-MEM algorithm [27] with default parameter settings. The mapped read files (*.sam) were subsequently converted to sorted compressed binary version BAM files using SAMtools [28].

Where gene knockouts were validated using standard PCR, the Q5^®^ Hot Start High-Fidelity DNA Polymerase (NEB) and primers designed to target two different areas (Appendix A) were used. Internal primers targeted the gene while external primers targeted regions outside the gene. Hygromycin specific primers were also designed. The PCR thermocycling conditions for the Q5 polymerase were followed, but with an initial denaturation of 2:00 min.

### 2.6. Confrontation Plate Assays of C. rosea and F. graminearum

PDA plates were labelled, and two lines were drawn 6 cm apart for separate fungal inoculations (Figure 1A). Spores were thawed quickly and diluted in sterile water to a concentration of 12,000 spores/mL. The diluted spores were dispensed into reservoirs. The *C. rosea* spores were stamped onto one line on the PDA plate, using a 50 × 50 mm sterile glass microscope slide and allowed to dry onto the plate. These plates were kept at 25 °C in a windowless incubator for 3–4 days, to allow *C. rosea* growth to establish (Figure 1B). *F. graminearum* spores were extracted fresh, 1–2 days prior to inoculation on the agar and diluted to 12,000 spores/mL concentration. On day 3 or 4, the Fg spores were stamped on the agar, on the opposing line and both fungi grew at 25 °C until day 7, where the two fungi interacted (Figure 1C,D). Using sterile sampling straws, 12 agar plugs were collected from the interaction zone across diameter of the plate center into borosilicate scintillation vials—A total of 6 replicates per strain were performed (Figure 1E). The agar plugs were extracted with 10 mL of ethyl acetate (Fisher, E154), shaking at 100 rpm for two hours and the resulting solvent extract was then dried down using the EZ-2 Elite Personal Evaporator (GeneVac^®^) at setting 2: Low BP at 40 °C for approximately 20 to 30 min. The dried extract was resuspended in 0.5 mL HPLC grade methanol (Fisher, A454) for UPLC-HRMS analysis.

### 2.7. High-Resolution Mass Spectrometry (HRMS) Analysis

The extracts were analyzed using a Thermo Ultimate 3000 UPLC coupled to a Thermo LTQ Orbitrap XL high-resolution mass spectrometer (HRMS) and chromatography was performed on a Phenomenex Kinetex 1.7 µm C_18_ column (50 × 2.1 mm, 100 Å). A mobile phase consisting of H_2_O + 0.1% formic acid (solvent A) and ACN + 0.1% formic acid (solvent B) was used at a flow rate of 0.35 mL/min. For each injection the column was held at 5% B for 0.5 min, increased to 95% B over 4.5 min, remained at 95% B for 3.5 min, then returned to 5% B over 1 min and was left to equilibrate for 3 min before the next injection. Positive mode electrospray ionization (ESI^+^) was used to ionize the eluent. The HRMS was operated at a resolution of 30,000 and scanned a range of 100–2000 *m/z* with the following parameters: sheath gas flow 40, auxiliary gas flow 5, sweep gas flow 2, spray voltage 4.2 kV, capillary temperature 320 °C, capillary voltage 35 V, tube lens 100 V, AGC target 5E5 and maximum ion time 500 ms. Samples were injected in a randomized order and MeOH blanks were injected after every 10 samples to evaluate metabolite carry over. Peak detection and integration of 15-ADON and 15-ADON-3G was achieved using Xcalibur Software (Thermo Scientific). Peak areas were exported into an Excel spreadsheet and the peak areas of 15-ADON-3G were normalized by that of 15-ADON in each sample. Univariate statistical analyses (Student’s T test and Mann-Whitney test) were performed to assess differences in 15-ADON-3G relative abundance in solvent extracts between *C. rosea* WT and UDP-GTase knockout strains.

### 2.8. Differentially Expressed Gene Analysis

Differentially expressed gene data were obtained from RNAseq data previously collected for a plate confrontation assay between *C. rosea* ACM941 and *F. graminearum* GZ3639, published by Demissie et al. [19].

## 3. Results

### 3.1. In Silico Data Mining and Structural Comparisons of UDP-GTases 

The string search of the *C. rosea* CBS125111 v1.0 genome in the JGI database yielded 50 sequences, of which 19 were identified as belonging to UDP-glycosyltransferase (UGT) Family 1 (EC 2.4.1.x;) (Appendix A). Two additional sequences were identified as α, α-trehalose-phosphate synthase [UDP-forming] and were screened for a total of 21 possible targets going forward. Toward further reducing the number of candidates, amino acid sequence alignments were performed of the identified targets with other known UDP-GTases that act on DON. These revealed identities to the DON glycosylating rice Os79 UDP-GTase for example, ranging from 15–24%, which was not surprising for a plant vs. fungus comparison, but also not useful for screening targets out. 

Thus, we applied a protein structural comparison strategy. Toward this, the 21 amino acid sequences were used as input for the creation of 3-dimensional structural homology models. Comparative visual inspection of the models with respect to available crystal structures of either rice Os79 UGT (PDBID # 5TMD) and *Arabidopsis* UGT51 (PDBID # 5GL5) highlighted similar overall folds but an array of variations in the conservation of catalytic and ligand binding residues at the overlapping positions in the active site, that were not readily detectable based on sequence alignments (Figure 2; Appendix A). The structure of PDB ID # 5GL5 was selected for comparison because the Swiss model uses this structure as the template for many of the homology models (Appendix A). Ligand binding and catalytic residues were defined based on the inspection of the rice (Os70 PDB ID # 5TMD) and *Arabidopsis* (UGT51 PDB ID # 5GL5) structures, which were previously co-crystallized with a fluorinated uridine-diphosphate-glucose analogue and 12,13-epoxytrichothec-9-ene (EPT) or just uridine-diphosphate-glucose, respectively [16,18,29]. In the case of the rice Os79 enzyme, the model is predicted to rely on a His27-Asp120 catalytic dyad to activate the ETP O3 hydroxyl for the nucleophilic attack at the C1′ of the UDP-glucose donor (Figure 2A). In contrast, *Arabidopsis* UGT51 presents a catalytic Asp (Asp752), in place of the rice His27, for the activation of the hydroxyl group of a substrate for the nucleophilic attack at the C1′ of the UDP-glucose donor (Figure 2B). 

Inspection of the catalytic active site of the CrUGT1 and CrUGT3 models showed that both models had a conserved His-Asp catalytic dyad similar to that observed in Os70 (Figure 1C,D). Interestingly, overlay of the CrUGT4 model with Os79 shows that the longer side chain of E141 in CrUGT4 forces the residue into a potentially non-catalytic conformation compared to D120 in Os70 (Figure 1E). In contrast, overlay of the CrUGT5 model (pink) with the *Arabidopsis* UGT51 structure (white), showed that the Asn substitution for Ser at position 847 of Os79 is likely not detrimental to activity (Figure 1F). On this basis, ten targets that conserved either the rice catalytic dyad or the *Arabidopsis* catalytic Asp were selected for functional characterization (Table 1).

Finally, we considered the possibility of two α, α -trehalose-phosphate synthases that were annotated as UDP-generating and referred to as CrUTPS1 and CrUTPS2 herein (Appendix A). Much like the Family 1 UDP-GTases, these synthases transfer sugar moieties from UDP to selected metabolites but rely on a different enzymatic mechanism that is not strictly associated with conserved catalytic amino acid residues. Rather, the synthases rely on the unique placement/orientation of the substrates in proximity to each other in the active site, such that elements of the substrates themselves catalyze the reaction [30,31,32]. Structural analysis of the CrUTPS2 active site suggests the active site pocket may accommodate both the 15-ADON and UDP substrates, while a potential clash between the large Arg 424 side chain in CrUTPS1 and the uridine moiety of UDP might be prohibitive (Appendix A). The equivalent residue in CrUTPS2 is a Val 284, which is more spatially/electrochemically accommodating.

### 3.2. Functional Evaluation of Select UDP-GTase Deletions on 15-ADON-3G Production by C. rosea

We attempted to knockout 12 different *C. rosea* UDP-GTases using a CRISPR/Cas9 strategy (Table 1). Of the 12 targets, 9 were successfully replaced with a gene encoding for hygromycin including CrUGTs 1–3, 6–9 and CrUTPSs 1 and 2. Deletions were confirmed either by PCR and/or by whole genome sequencing (Appendix A). Obtained *C. rosea* UDP-GTase knockout strains were subsequently tested for their ability to detoxify 15-ADON by glycosylation in confrontation plate assays against *F. graminearum* (Figure 3; Appendix A). UPLC-HRMS profiling of extracts derived from agar plugs taken from the interaction zone demonstrated some degree of reduction in the accumulation of 15-ADON-3G between most UDP-GTase knockout strains compared to WT (with the exception of CrUGT2 and CrUGT8). Three of the knockout strains (CrUGT3, CrUGT6 and CrUGT9) had a statistically significant reduction in 15-ADON-3G compared to the WT. Interestingly, complete abolishment of 15-ADON-3G production was not observed in any of the UDP-GTase knockout strains examined.

### 3.3. Differential Gene Expression Evaluation for Selected UDP-GTases

In order to assess the impact of the targeted KO UDP-GTases on gene expression in wild-type *C. rosea* during confrontation with *F. graminearum*, previously obtained differentially expressed gene data arising from a plate confrontation assay between *C. rosea* ACM941 and *F. graminearum* GZ3639, compared to *C. rosea* ACM941 alone, was mined [19]. Of the ten UDP-GTases that were predicted to maintain the rice catalytic dyad or the *Arabidopsis* catalytic Asp catalytic signatures, eight were found to have transcript homologs in *C. rosea* IK726 that were matched to differentially regulated gene homologs in *C. rosea* ACM941 (Figure 4). Of these eight, three were assessed as being associated with 15-ADON-3G production by KO including CrUGT3, CrUGT6 and CrUGT9 (Figure 3). CrUGT3 and CrUGT9 showed only limited differential gene induction in the presence of *F. graminerum*, with log2fold change values 1.5 > X < 2.0, and with only the CrUGT9 difference being deemed significant. In contrast, CrUGT6 showed a significant 4.5-fold induction in expression in the presence of *F. graminearum*. The most highly upregulated target, CrUGT7, showed only limited production of 15-ADON-3G and we also note that this hit shared only an 89% identity with the IK726 transcript homolog.

## 4. Discussion

UDP-GTases form a large family of enzymes involved in the transfer of sugars from UDP to a broad array of different metabolites [33,34]. Different UDP-GTases target structurally different metabolites, with many UDP-GTases being able to function against more than one metabolite and the majority of predicted UDP-GTases currently being uncharacterized. In this report, in silico modelling of the structural characteristics of an array of *C. rosea* UDP-GTases was carried out to predict the enzymes that might putatively contribute to glycosylation, and thus the detoxification of the *F. graminearum* mycotoxin DON.

With respect to enzymatic targets that maintained the Os70-like catalytic dyad and other Os70 active site attributes [16,18], notably, the knockout of CrUGT3 led to a significant decrease in the accumulation of 15-ADON-3G (Figure 3). This is entirely consistent with its almost complete conservation of the Os70 catalytic active site residues (Table 1; Figure 2D) despite only 18% of the amino acid sequence identity being shared between Os70 and CrUGT3 overall. The fact that the Thr 19 residue in CrUGT1 changed from Gly 26 in Os70, does not preclude functionality despite its location immediately proximal to the catalytic site being notable. The role of Thr 19 in substrate specificity and reaction selectivity remains to be evaluated in future, more detailed, structure and function studies. The observed activity of CrUGT3 is also consistent with observed *F. graminearum*-induced upregulation of gene expression levels of a gene in strain AMC941 that associates with the IK726 transcript BN869_T00005396_1 which is 95% identical to CrUGT3 (Figure 4; (*p* = 0.06)). Thus, it is likely that CrUGT3 makes contributions to the mycoparasitic activity of *C. rosea* and DON detoxification. 

In this vein, it is then interesting that knockout of CrUGT2 did not show any effect on the accumulation of 15-ADON-3G, being that its active site residues are completely conserved with CrUGT3 (Table 1). Possible explanations include that its amino acid identity to Os70 is even lower than CrUGT3 at 17%, and it only shared 48% identity to CrUGT3. Thus, other residues more distal from the active site could be impacting its functionality. However, consideration of the DEG data highlights that the most likely explanation for the observed lack of effect is that there is no detectable induced expression of any gene with homology to CrUUGT2 in *C. rosea,* upon confrontation with *F. graminearum* (Figure 4). Thus, even if CrUGT2 has the ability to glycosylate 15-ADON, the knockout of CrUGT2 would not be predicted to have any detectable effect on 15-ADON-3G production, consistent with observations herein. Similarly, we also note only minimal evidence of expression of any *C. rosea* gene with homology to CrUGT1 in the presence of *F. graminearum* (Figure 4), such that whether the noted amino acid differences in the active site have functional consequences can not be assessed here (Table 1; Figure 2E).

Two additional hits, showing linkage to 15-ADON-3G accumulation were obtained from the collection of CrUGTs that conserve the *Arabidopsis* UGT51 active site residues. These include CrUGT6 and CrUGT9, who only share 27% amino acid sequence identity, but display an identical active site amino acid profile to each other (Table 1; Figure 2F). The observed activity of CrUGT6 and CrUGT9 are also consistent with observed *F. graminearum*-induced upregulation of gene expression levels of two genes in strain AMC941 that associates with the IK726 transcripts BN869_T00004378_1 and BN869_T00006025_1 (Figure 4; (*p* = 0.06)). These IK726 genes are 92 and 93% identical to CrUGT6 and CrUGT9, respectively. The observed gene sequence similarity and upregulated expression during *C. rosea*/*F. graminearum* confrontation supports a proposed role for these two enzymes in mycoparasitism and DON detoxification. Notably both CrUGT6 and CrUGT9 share 33% and 51% amino acid sequence identity to the *Arabidopsis* UGT51, where UGT51 has not been linked to DON detoxification [29]. The *C. rosea* targets incorporate one change in the active site residue profile, where Ser 847 in *Arabidopsis thaliana* is an Asn in *C. rosea*. Unfortunately, whether this change is significant to functionality remains enigmatic, due to the inability to obtain a knockout of the CrUGT5 target, which was the only target to maintain 100% active site conservation compared to UGT51 (Table 1).

Interestingly, CrUGT7 or CrUGT8 knockouts showed little impact on 15-ADON-3G accumulation, despite 100% amino acid active profile conservation with CrUGT6 and CrUGT9. This is particularly notably for CrUGT7, which shares homology with an IK726 transcript associated with an ACM941 gene that had the most highly induced gene expression of any of the targets upon confrontation with *F. graminaerum* (Figure 4). However, at only 89% sequence identity to the IK726 transcript, it is possible that the regulated ACM941 gene is not a representative of CrUGT7, or that other amino acid changes distal from the active site impact activity against 15-ADON negatively. A similar interpretation of the CrUGT8 data applies here. Finally, although some expression of an ACM941 differentially expressed gene was detected for CrUGT10, relevance could not be assessed due to the inability to obtain a successful knockout of this target. Notably, only a single CRISPR/Cas9 knockout attempt was made for each target, due to the higher number of targets being assessed (i.e. a high-throughput strategy was applied with negatives triaged in the first round). With some tweaking of the CRISPR/Cas9 constructs, knockouts of the other targets might be obtained in the future.

In conclusion, we successfully identified three novel fungal *C. rosea* UDP-GTases (CrUGT3, CrUGT6 and CrUGT9) validated to contribute to the glycosylation of 15-ADON to produce 15-ADON-3G during confrontation with *F. graminearum*. While CrUGT3 maintains a similar catalytic site to other known plant UDP-GTases with documented activity against mycotoxins, it is notable that CrUGT6 and CrUGT9 do not. Instead, CrUGT6 and CrUGT9 more closely approximate a known sterol glycosyltransferase from *Arabidopsis*, highlighting the broad potential substrate specificity of members of the UDP-GTase family. Other identified UDP-GTases (CrUGT2, CrUGT7 and CrUGT8) may also have the capacity to glycosylate 15-ADON, based on structural organization of their active sites, but do not appear to be relevant in the biological context of the plate confrontation assay against *F. graminearum* used in this study. At the same time, the possibility that the single knockouts may have induced compensatory expression of other enzymes, contributing to the observations described herein, remains for future evaluation.

## Figures and Tables

**Figure 1 jof-09-00723-f001:**
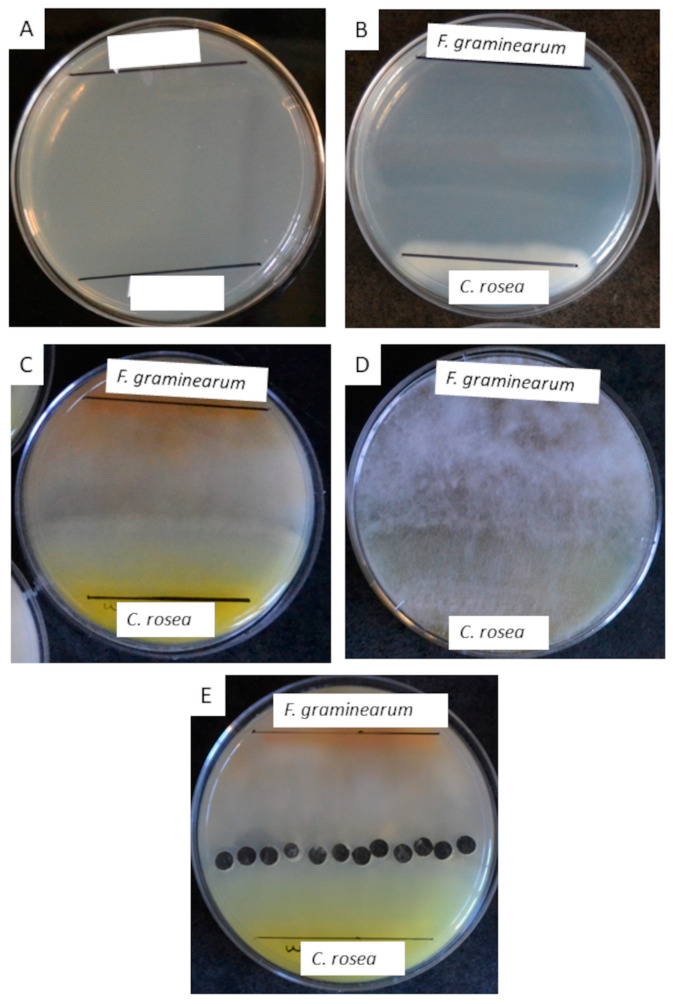
Details of Confrontation Assay between *C. rosea* and *F. graminearum.* (**A**) A PDA plate is prepared with lines drawn 6 cm apart. (**B**) *C. rosea* spores were stamped onto one line on the PDA plate, using a 50 × 50 mm sterile glass microscope slide and allowed to dry into the plate. These plates were kept at 25 °C in a windowless incubator for 3–4 days, to allow *C. rosea* growth to establish. (**C**) *F. graminearum* spores were stamped on the agar, on the opposing line and both fungi grown at 25 °C until day 7. (**D**) Same plate as in C but flipped over to view the other side. (**E**) Using sterile sampling straws, 12 agar plugs were collected from the interaction zone across the plate for ethyl acetate extraction.

**Figure 2 jof-09-00723-f002:**
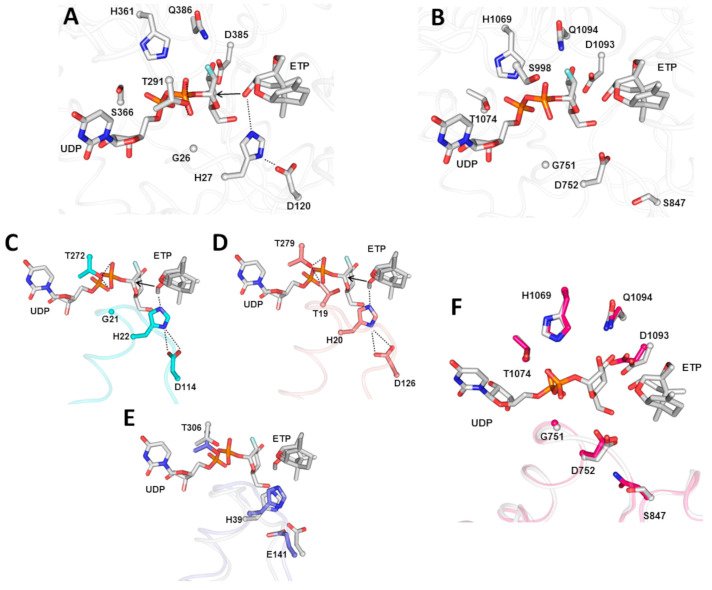
Identification of the ligand binding and catalytic sites for two UDP-GTases of known structures. Structures include I, (**A**) Oryza sativa UDP-GTase Os79 from PDB ID # 5TMD and (**B**) Saccharomyces cerevisiae UDP-Gtase UGT51 from PDB ID # 5GL5. Amino acid side chains involved in coordinating the binding of the UDP and the 12,13- epoxytrichothec-9-ene (EPT) analogue of the mycotoxin DON are shown. The catalytic dyad in Os79 is indicated with dashed lines. (**C**) Catalytic active site of the CrUGT1 model showing conservation of the His22-Asp114 catalytic dyad seen in Os70. (**D**) Catalytic active site of the CrUGT3 model also showing conservation of the His20-Asp126 catalytic dyad consistent with Os70. The model of CrUGT2 was identical to CrUGT3 as expected (see Table 1). (**E**) Overlay of the CrUGT4 model (blue) with Os79 (white) showing that the longer side chain of E141 forces the residue into a potentially non-catalytic conformation compared to D120 in Os70. (**F**) Overlay of the CrUGT6 model (pink) with the Arabidopsis UGT51 structure (white), showing that the Asn substitution for Ser at position 847 of Os79 is likely not detrimental to activity. For all of B-F the UDP and ETP analogues are shown based on structural alignment with the 5TMD structure of Os70.

**Figure 3 jof-09-00723-f003:**
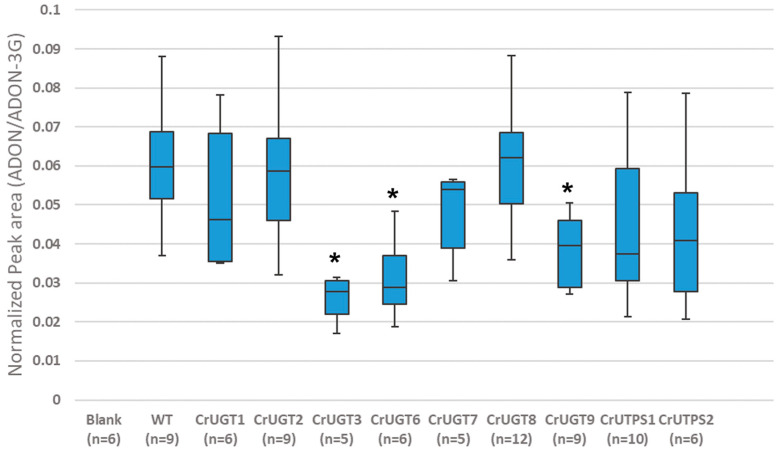
Boxplot representation of the normalized peak area of 15-ADON-3G detected in extracts of UDP-glycotransferase KO strains. The number of samples analyzed for each strain is indicated in brackets (n). Significance was calculated by student-T and Mann Whitney tests (Appendix A). Samples with *p*-values < 0.005 in both statistical tests are indicated with a *.

**Figure 4 jof-09-00723-f004:**
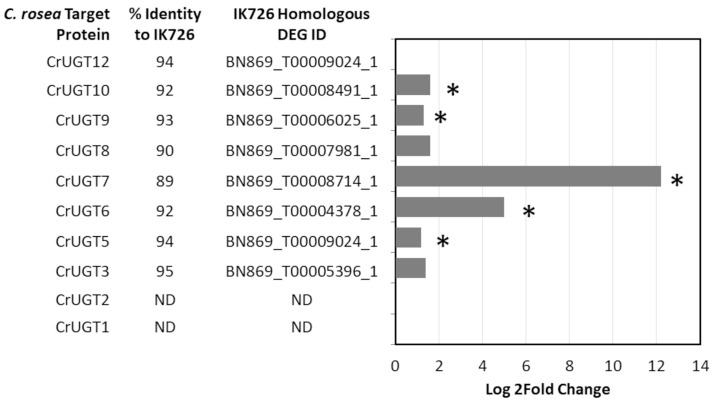
Differentially Expressed Gene Analysis. Knockout targets in this study that had an identifiable *C. rosea* IK726 homolog that was in turn homologous to the differentially expressed *C. rosea* ACM941 genes in a previous confrontation experiment between *C. rosea* strain ACM941 and *F. graminearum* [19]. Log2Fold Change values with *p*-values < 0.05 are indicated with an *.

**Table 1 jof-09-00723-t001:** Putative UDP-glycosyltransferase targets selected for validation by knockout in *C. rosea* based on comparative structural alignments *.

***C. rosea* CBS125111v1.0 jgi Protein ID (From Appendix A)**	**Assigned Protein Name**	**Active Site Amino Acid Resides (Number or Type)**								**Modelling Template** **(RCSB PDB ID)**	**% Identity of *C. rosea* CBS125111v1.0 Protein to ACM941 RNAseq Data**	***C. rosea* ACM941 Fasta Entry [19] ****	***C. rosea* ACM941 Gene ID *****
	Os70	**26**	**27**	**120**	**291**	**361**	**366**	**385**	**386**				
	UGT51	**751**	**752**	**847**	**998**	**1069**	**1074**	**1093**	**1094**				
	Os70	**G**	** H **	** D **	**T**	**H**	**S**	**D**	**Q**	**5TMD**			
	UGT51	**G**	** D **	**S**	**S**	**H**	**T**	**D**	**Q**	**5GL5**			
493176	**CrUGT1**	**G**	**H**	**D**	**T**	**N**	**S**	**D**	**K**	**2IYA**	**93**	JACYFL010000055.1	scf_055.r.213883
501281	**CrUGT2**	**T**	**H**	**D**	**T**	**H**	**S**	**D**	**Q**	**6LNF**	**94**	JACYFL010000025.1	scf_025.r.655870
472238	**CrUGT3**	**T**	**H**	**D**	**T**	**H**	**S**	**D**	**Q**	**5V2K**	**93**	JACYFL010000016.1	scf_016.r.712528
493176	CrUGT4	**G**	**H**	**E**	**T**	**N**	**G**	**D**	**H**	**2IYA**	**94**	JACYFL010000472.1	scf_472.r.2070
565562	CrUGT5	**G**	**D**	**S**	**S**	**H**	**T**	**D**	**Q**	**5XVM**	**91**	JACYFL010000004.1	scf_004.f.484466
451096	**CrUGT6**	**G**	**D**	**N**	**S**	**H**	**T**	**D**	**Q**	**5GL5**	**91**	JACYFL010000010.1	scf_010.r.539615
497403	**CrUGT7**	**G**	**D**	**N**	**S**	**H**	**T**	**D**	**Q**	**5GL5**	**94**	JACYFL010000005.1	scf_005.f.859807
473884	**CrUGT8**	**G**	**D**	**N**	**S**	**H**	**T**	**D**	**Q**	**1PNV/5Gl5**	**93**	JACYFL010000030.1	scf_030.f.8134
597105	**CrUGT9**	**G**	**D**	**N**	**S**	**H**	**T**	**D**	**Q**	**5GL5**	**92**	JACYFL010000032.1	scf_032.r.200735
257207	CrUGT10	**G**	**D**	**N**	**S**	**H**	**T**	**D**	**Q**	**5GL5**	**94**	JACYFL010000013.1	scf_013.r.885225
301967	**CrUTPS1**									5HUT	99	JACYFL010000005.1	scf_005.r.669569
409634	**CrUTPS2**									5HUT	97	JACYFL010000030.1	scf_030.f.441167

* Os70 and UGT51 catalytic resides shown in red font. Green shading refers to Os70 or Os70-like; blue shading refers to UGT51 or UGT51-like; grey shading indicates not identical to either Os70 or UGT51; white shading indicates identical in/to both Os70 and UFT51. Bolded assigned protein names highlight targets that were successfully knocked out. ** Genes found in the JGI database (Appendix A) were aligned to the shotgun sequences of ACM941 [19] yielding the fasta entry that gene was found in. *** The last 3 digits of the fasta entry are the first numbers in the AMC941 gene ID code. The ‘r’ or ‘f’ indicates if the gene was in the forward or reverse direction. The last number corresponds to the index where the gene starts.

## Data Availability

The RNAseq data presented in this study are available in [19].

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
