# Peer review of "Multiple Clonostachys rosea UDP-Glycosyltransferases Contribute to the Production of 15-Acetyl-Deoxynivalenol-3-O-Glycoside When Confronted with Fusarium graminearum"

_jof, 2023, doi:10.3390/jof9070723_

Round 1

Reviewer 1 Report

This paper showed that conversion of 15-acetyl-DON (15-ADON) into 15-ADON-3-O-glycoside (15-ADON-3G) in Clonostachys rosea.carried out by a uridine diphosphate glycosyl transferase (UDP-GTase).It was proved by leveraging the availability of the crystal structure of rice Os70, and revealed multiple UDP-GTase enzymes each contributing partial activities. It is useful for clear the tranfroamtion and metabolismof DON and its derivate. 

Some suggestions, 1.  in the reference, some journal name is abbreviation while others are not, please be consisitant.

2. The significant difference in the Figure was noted by *,it would be better if they were showed in different letters

Author Response

  1. in the reference, some journal name is abbreviation while others are not, please be consisitant.

RESPONSE 1: All reference journal names are now abbreviated according to the international standard for journal abbreviations.

  1. The significant difference in the Figure was noted by *,it would be better if they were showed in different letters

RESPONSE 2: Because the p-values across the two different tests for the KO strains that were deemed significantly different from wild-type are so similar(p-values ranges from 0.0034 to 0.0001) we don’t believe there is any need to differentiate them, especially considering the detailed p-values for each KO are included in Supplemental Table S6 in Supplemental File 1.

Reviewer 2 Report

This paper first identified a subset of potential C. rosea UDP-GTases that might have activity against 15-ADON by using protein structural comparison strategy based on the crystal structure of rice Os70. Subsequently, this study took advantage of CRISPR/Cas9 technology to knock out target UDP-GTase in C. rosea strain ACM941 and evaluate the production of 15-ADON-3G in confrontation assays with F. graminearum to determine the major UDP-GTase enzymes contributing to detoxification activities. In a word, this test further explored the previous research, which was innovative and provided a reference for analyzing the detoxification process of DON in C. rosea strain.

However, some minor issues still need to be explained and improved:

1. Why are the number of samples analyzed different for different treatment groups in Figure 3?

2. The Figure 3 is not aesthetically pleasing or intuitive, and we recommend using a columnar scatter plot to present the results.

3. The notes about the contents of a table or figure should be placed uniformly below it.

4. In this paper, CRISPR/Cas9 knockout of a single UDP-GTase was used to identify major functional enzymes. Whether this might induce compensatory expression of other enzymes and affect the rigor of the conclusions. 

The phrasing and grammar of this article are quite standard.

Author Response

  1. Why are the number of samples analyzed different for different treatment groups in Figure 3?  RESPONSE 1: Each knockout strain (treatment group) was evaluated by taking 5-6 different extractions from confrontation plates. In some cases, but not all, we had additional confrontations plates and extracted an additional 5-6 samples for those conditions. We considered limiting the data to only 5-6 extraction/condition, but decided in the end to publish everything we had collected.  The additional 5-6 samples/ condition had little impact on the overall observed averages and statistics.
  1. The Figure 3 is not aesthetically pleasing or intuitive, and we recommend using a columnar scatter plot to present the results. RESPONSE 2: Agreed!  Plot has been replotted as a Boxplot. Thank you for the suggestion.
  1. The notes about the contents of a table or figure should be placed uniformly below it. RESPONSE 3: All notes for Table 1 have been placed below it.  I believe infact the Title of Tables should remain above and have left it there accordingly.  All Figure titles and notes were already consistently placed below each figure.
  2. In this paper, CRISPR/Cas9 knockout of a single UDP-GTase was used to identify major functional enzymes. Whether this might induce compensatory expression of other enzymes and affect the rigor of the conclusions.  RESPONSE 4: We appreciate the reviewer’s comment and have added a sentence addressing this issue in the conclusions lines 467-469.

Reviewer 3 Report

The manuscript studied C. rosea UGTs in relation to the production of 15-ADON-3-glycoside from Fusarium graminearum. The strategy combined bioinformatics/protein modeling and CRISPR-Cas9 technology to identify bona fide CrUGTs that transform 15-ADON to 15-ADON-3G. 

The study was well designed and conclusions were coherently supported on basis of the results.

A minor comment is whether the authors can postulate the reason CrUGT4/5/10 escaped gene editing. 

Author Response

A minor comment is whether the authors can postulate the reason CrUGT4/5/10 escaped gene editing. 

RESPONSE: We only made a single attempt at Crispr KO (ie single constructs), due to the higher number of targets we had, and thus chose to simply triage those that did not yield a KO in the first round (ie we applied a high-throughput strategy).  With some tweaking of the crispr constructs we might be able to obtain the KOs in question in the future. I have added this idea to the discussion lines 452-456 in the revised version highlighted in read.